# The Affective Responses to Moderate Physical Activity: A Further Study to Prove the Convergent and the Discriminant Validity for the German Versions of the Feeling Scale and the Felt Arousal Scale

**DOI:** 10.3390/bs14040317

**Published:** 2024-04-12

**Authors:** Kristin Thorenz, Gorden Sudeck, Andre Berwinkel, Matthias Weigelt

**Affiliations:** 1Department of Sport and Health, University of Paderborn, 33098 Paderborn, Germany; 2Institute of Sports Science, University of Tübingen, 72074 Tübingen, Germany; 3University Clinic for Psychiatry and Psychotherapy (EvKB), Campus Bielefeld-Bethel, University of Bielefeld, 33617 Bielefeld, Germany

**Keywords:** Feeling Scale, Felt Arousal Scale, physical activity, affective responses

## Abstract

The present study proves the construct validity of the German versions of the Feeling Scale (FS) and the Felt Arousal Scale (FAS) for measuring the affective responses (affective valence and arousal) for a moderate-intensity jogging (JG) exercise. In previous studies, both scales were validated for a high-intensity bicycle ergometer exercise and for relaxation techniques. In the present study, 194 participants performed the JG exercise for 45 min and completed the FS and the FAS, as well as the Self-Assessment Manikin (SAM), for a self–other comparison in a pre-test-intervention-post-test design. The results of the correlation analyses replicated the previous findings for the high-intensity bicycle ergometer exercise and the relaxation techniques, revealing significant positive correlations for the valence dimension between the FS and the SAM-Pleasure subscale (*r* = 0.50) and for the arousal dimension between the FAS and the SAM-Arousal subscale (*r* = 0.16). These findings suggest that the German versions of the FS and the FAS are also suitable for exercises of moderate intensity.

## 1. Introduction

Regular physical activity benefits psychological well-being and mental health [1]. In this regard, physical activity of moderate intensity for 150 min or of vigorous intensity for 75 min per week and muscle strength training on two or more days a week is recommended by the World Health Organization [2]. Unfortunately, it is not enough that people are aware about health benefits from regular physical activity for behavioral changes to occur. Rather, this awareness must be supplemented by a positive association with the exercise experience [3]. Brand and Ekkekakis [4] explain this mechanism using affective-reflective theory (ART), where the affective responses following physical activity (e.g., pleasure or displeasure, tension or relaxation) play an important role in behavioral change [5,6]. According to Lee et al. [7], the personality (traits) of exercisers and the individually perceived benefits of leisure time activities also play a role in whether a physical activity is performed or repeated. Furthermore, Sudeck and Thiel [8] emphasize that the same physical activity may have different effects on an individual level, and therefore, when two or more people are engaged in the same exercise, their affective responses will differ from person to person. In addition to personal factors, the intensity of physical activity is a moderator for the affective responses [9]. Evaluation of various meta-analyses shows, on the one hand, a significant relationship between high-intensity exercises and an increase in unpleasant valence and, on the other hand, more pleasant valence after a single bout of low-intensity exercise compared to moderate- and high-intensity exercises [6,8]. In addition, Brehm and Pahmeier [10] pointed out that a positive emotional association with an exercise-based intervention promotes people’s adherence to physical activity (see also [4]). Providing direct feedback to quantify the subjective experience after physical exercise is therefore especially important for inactive people in order for them to associate the (positive) affective responses with the exercise. 

Therefore, the goal should be to find the optimal level of exercise intensity for each individual exerciser and not to recommend an intensity level that everyone should feel good with. As an indicator for the optimal level of exercise intensity, the ventilatory threshold (VT) can be used; this differentiates the aerobic and anaerobic energy-producing pathways to supply the muscles. Exercising above the VT (i.e., while relying predominantly on anaerobic energy supply) leads to an increase in unpleasant feelings [11,12,13]. Especially, studies on high-intensity interval training (HIIT) show a clear link between homeostatic disturbances and affective valence. That is, during the high-intensity interval of the HIIT, the perceived affective valence decreases, while it increases again when the exercise is interrupted and after the completion of the whole exercise (the so-called rebound effect) [14,15]. Thus, the VT also separates the ongoing evaluation of the exercise effect into inducing pleasant (below the VT) and unpleasant (above the VT) feelings [13]. It follows that the VT can be used to find the optimal intensity or to customize training regimens so that the intensity is perceived as pleasant [14]. The affective valence can be assessed by the single-item Feeling Scale (FS; [16]) questionnaire during exercise. 

Another important factor in finding the optimal intensity is to choose a self-selected intensity. Studies have shown that participants increased their intensity (around the VT) over the duration of exercise to maintain a pleasant feeling [13,17,18]. This adaptation requires cognitive appraisal processes and results in reduced response variability across participants in terms of affective valence during the exercises [13,18]. In this regard, the dual-mode model (DMM) of Ekkekakis [19] describes the role of the VT and the cognitive appraisal process in affective responding for physical activity. As the intensity around the VT increases, cognitive appraisal becomes more important; i.e., the individual person assesses whether the effort is challenging but manageable (more pleasant) or whether the effort is beyond one’s abilities and poses a risk to health (unpleasant). The highest level of effort described is defined as heavy intensity [20]. If the intensity increases above the VT, the dominant processing of interoceptive cues (e.g., heart rate, respiration) leads to unpleasant feelings. 

In addition to the affective valence, the affective arousal is important for examining affective responses to physical exercise [21]. The Felt Arousal Scale (FAS, [22]) is recommended for assessing arousal [5]. The FAS is a single-item scale and one part of the Telic State Measure (TSM; [22]); it measures how aroused participants currently feel. The assessment of change in affective responses to physical activity should be simple, economical, and applicable to different types of exercise and settings. Ekkekakis and Petruzello [5] see these advantages in the two single-item scales, the FS and the FAS. Therefore, and following the circumplex model by Russell [23], the researchers recommend crossing these two scales. Accordingly, four quadrants are formed: unactivated-pleasant affect, unactivated-unpleasant affect, activated-unpleasant affect, and activated-pleasant affect [5]; this model offers the possibility to predict various affective states within a two-dimensional model (e.g., tension, energy, relaxation, boredom) depending on one’s own perceived valence and activation (arousal). Therefore, exercising at a high-intensity level that exceeds the individual VT of a person will lead to an activated-unpleasant affective state of distress or tension, while, in contrast, exercising at a low-intensity level below the VT will result in an unactivated-pleasant affective state of calmness or relaxation.

Especially in German-speaking countries, however, multidimensional measurements with multiple items are used to assess actual well-being in relation to physical activity; e.g., the “Befindlichkeitsskala”, by Abele-Brehm and Brehm [24], includes 40 items to be rated by the subjects, and these are assigned to eight dimensions (e.g., calmness, anger arousal); the Multidimensional Mood Questionnaire (MDMQ) by Steyer et al. [25] uses 32 items, or 12 items for the short form [26], assigned to three bipolar dimensions (feeling well vs. feeling bad, being awake vs. being tired, feeling calm vs. feeling tense); or an adapted version of the MDMQ by Wilhelm and Schoebi [27], in which six bipolar items are assessed for the three dimensions as described for the MDMQ. With the two single-item Feeling Scale (FS; [16]) and Felt Arousal Scale (FAS; [22]) questionnaires, the measurement instruments for the research field of affective responses in German-speaking countries can be extended by valid measurement instruments that capture the basic structure of emotional states (affective valence and arousal) and enable an international comparison. 

Recently, Maibach et al. [28] translated the FS and the FAS into German and validated the two scales for a high-intensity bicycle ergometer task. A total of 82 participants completed the exercise with a gradual increase in physical load up to subjective exhaustion. The affective responses were measured at different times during the exercise with the FS and the FAS, as well as with the Self-Assessment Manikin (SAM; [29]). The SAM is a non-verbal measurement for assessing affective states based on three affective dimensions: pleasure, arousal, and dominance. To capture the subjective feeling of effort, the rating of perceived exertion (RPE; [30]) was also used. With the correlation analyses between the FS and the subscale pleasure (P) of the SAM (*r* = 0.72 to 0.73) and the FAS and the subscale arousal (A) of the SAM (*r* = 0.50 to 0.62), the results for high-intensity exercise (i.e., the bicycle ergometer task) were comparable to the English validation study [28]. Also, two previous studies proved the construct validity for the FS and the FAS for two relaxation techniques: progressive muscle relaxation (PMR; [31]) and autogenic training (AT; [32]). Relaxation techniques are clearly different from high-intensity exercise in terms of energy-producing pathways to supply the muscles (aerobic vs. anaerobic). A total of 228 sport science students participated in the validation study for the PMR exercise. For convergent validity, the correlation analysis revealed significant results between the FS and the subscale SAM-P for the valence dimension (*r* = 0.67) and between the FAS and the subscale SAM-A for the arousal dimension (*r* = 0.31). In total, 224 sports science students took part in the validation study for the AT exercise. For the convergent validity, the results show positive significant correlations between the FS and SAM-P (*r* = 0.62) and between the FAS and SAM-A (*r* = 0.25).

### 1.1. Purpose and Hypotheses of the Present Study

The present study follows the recommendation for an ongoing construct validation for diagnostic tools (e.g., [33,34]). That is, whenever measurements are used in a new way, a new context, or for a new population, “evidence is needed to show that the scale scores are valid representation of the construct” [17], (p. 375). Therefore, the aim of the present study is to prove the construct validity of the German translations of the FS and the FAS by Maibach et al. [28] for a jogging (JG) exercise, which is a physical activity with a constant level of moderate intensity below or around the ventilatory threshold (VT). This kind of exercise was chosen to complement the high-intensity exercise of the bicycle ergometer task (above the VT) used by Maibach et al. [28] and the relaxation techniques below the VT examined by Thorenz et al. [31,32]. In addition, another aspect is the choice of the test population. While Maibach et al. [28] tested middle-aged adults (35–65 years), the present study examines young adults (18–35 years). To this end, participants were asked to run for 45 min at their self-selected, preferred speed, at which they felt pleasant. To assess potential changes of the affective valence and activation (arousal), participants completed the FS, FAS, and SAM before and after the JG exercise. Thereby, the SAM was administered for a self–other comparison to prove the construct validity of the single-item scale FS and the single-item scale FAS. 

### 1.2. Hypotheses for Correlation Analysis

For construct validity, if the German versions of the FS and the FAS prove to be also valid for the JG exercise performed at a constant level of moderate intensity, then a distinct pattern of correlations should be observable for the convergent validity and for the discriminant validity according to the criteria of multitrait-multimethod (MT-MM) analysis [35], respectively. That is, any two (sub)scales testing the same construct (FS and SAM-P, FAS and SAM-A) should be positively correlated with each other (indicating convergent validity, Hypothesis 1), while any two (sub)scales addressing different constructs (e.g., FS and SAM-A, FAS and SAM-P, FS and SAM-D) should either not be significantly correlated or show a negative correlation (supporting discriminant validity, Hypothesis 2).

**Hypothesis 1 (H1).** *Significant positive correlations between the FS and the SAM-P, and the FAS and the SAM-A are expected*.

**Hypothesis 2 (H2).** *No significant correlations between the FS and the SAM-A, and the FAS and the SAM-P are expected*.

### 1.3. Hypotheses for Descriptive Statistics

To compare the results of the descriptive statistics of the present study with the ones from the study by Maibach et al. [28] for a further validation, the means of the affective responses after the exercise (i.e., post-test values) will be considered for the FS and the FAS, respectively. For the JG exercise in the present study, a stronger affective response should be observed for the FS (as signaled by a higher mean post-test value, Hypothesis 3) and a weaker affective response for the FAS (as reflected by a lower mean post-test value, Hypothesis 4) than for the bicycle ergometer task in Maibach et al. [28] because of the different intensity levels of the two exercises (cf., [6,9]). That is, on average, participants should experience the moderate-intensity exercise in the present study as more pleasant and less strenuous than the participants who performed the high-intensity exercise in the study by Maibach et al. [28]. 

**Hypothesis 3 (H3).** *A higher mean post-test value for the FS is expected after the JG exercise than after the bicycle ergometer task*.

**Hypothesis 4 (H4).** *A lower mean post-test value for the FAS is expected after the JG exercise than after the bicycle ergometer task*.

### 1.4. Hypotheses for Changes in Affective Responses

Also, the magnitude and the direction of change in the affective responses from pre-test to post-test will be examined for the participants. Because of the (relatively) constant level of moderate intensity during the JG exercise in the present study, a smaller magnitude of change in the affective responses for both the FS (Hypothesis 5) and the FAS (Hypothesis 6) is expected as compared to the larger magnitude of change for the affective responses following the stepwise increase in physical load up to the level of subjective exhaustion during the bicycle ergometer task in Maibach et al. [28]. Also, the direction of change in the affective responses from pre-test to post-test should be positive for the FS and the FAS in the present study. In addition, the expected smaller magnitude of change for the FAS will be accompanied by more zero variations; i.e., more participants will perceive no changes in their arousal after the JG exercise as compared to the number of zero variations in Maibach et al. [28] (Hypothesis 7).

**Hypothesis 5 (H5).** *A smaller magnitude of change for the FS is expected for the JG exercise compared to the bicycle ergometer task*.

**Hypothesis 6 (H6).** *A smaller magnitude of change for the FAS is expected for the JG exercise compared to the bicycle ergometer task*.

**Hypothesis 7 (H7).** *A higher percentage of zero variations is expected for the FAS in the JG exercise than in the bicycle ergometer task*.

## 2. Materials and Methods

### 2.1. Participants

A total of 194 students (118 females and 76 males; age = 21.7 ± 2.1 years) participated in the study. All participants were sport science students at the University of Paderborn. The study took part in several bachelor courses on sport psychological training from the summer term in 2017 until the summer term in 2019. Participation in the study was voluntary; i.e., the successful completion of the course did not require students to register for the study. The registration for the study was anonymous, using a self-generated code which could not be traced back to the individual participant. The study was approved by the university’s local ethics committee.

### 2.2. Measurement

The affective responses to the different exercises were measured in a pre-test-intervention-post-test design, using paper-and-pencil tests. During the testing, three questionnaires were used to assess valence and arousal: the German translations of both the Feeling Scale and the Felt Arousal Scale [28] (see Appendix A) and the Self-Assessment Manikin [31].

Feeling Scale (FS; [28]): the FS is a numerical bipolar 11-point rating scale measuring the current mood. The odd numbers and zero are verbalized. The scale ranges from −5 (“very bad“), −3 (“bad“), −1 (“fairly bad“), 0 (“neutral“), +1 (“fairly good“), +3 (“good“), to +5 (“very good”) (German scale provided by Maibach et al. [28]; original scale from Hardy and Rejeski [16]).

Felt Arousal Scale (FAS; [28]): The FAS is a numerical 6-point rating scale measuring arousal from 1 (”low arousal”) to 6 (”high arousal”) (German scale provided by Maibach et al. [28]; original scale from Hardy and Rejeski [16]).

Self-Assessment Manikin (SAM; [29]): The SAM is a non-verbal pictorial assessment technique measuring affective state based on three major affective dimensions: pleasure (subscale SAM-P), arousal (subscale SAM-A), and dominance (subscale SAM-D). Five manikins are presented for each dimension. In the pleasure dimension (subscale SAM-P), the manikins range from happy smiling (5) to unhappy frowning (1). In the arousal dimension (subscale SAM-A), the manikins range from wide-eyed excitement (5) to sleepy relaxed (1). The dimension dominance (subscale SAM-D) represented five manikins of different sizes, from a small figure (1) to a large figure (5). The larger the figure, the more control is felt in the situation.

### 2.3. Procedure

In a first meeting, participants were provided with information about the study (procedure, data storage) and signed the informed consent form. In a second meeting, they were instructed to run for 45 min at a self-selected, preferred speed, at which they felt pleasant, without stopping or walking. In addition, they were advised to run without heavy breathing (in German “Laufen ohne zu schnaufen”) and in a way that they could still talk to each other during the run. Also, an experienced instructor was running with the participants, making sure that they would not run too fast (and instructing them to slow down if they did). The testing began with the participants completing the three questionnaires before the JG exercise (i.e., pre-test), followed by the JG exercise, and ending with the completion of the three questionnaires after the JG exercise (i.e., post-test). All questionnaires were presented as paper-and-pencil tests. The whole testing session lasted for about 55–60 min. The run was organized together with the seminar group and took place on running trails in the surroundings of the university, i.e., outdoors.

### 2.4. Data Analysis

A total of 206 sport science students participated in the JG exercise. Participants with missing values were excluded (12 participants). The data analysis was carried out with the data software IBM SPSS Statistics for Windows, version 29.0 (Armonk, NY, USA). To assess construct validity, the convergent validity (the pairs of the same construct) and the discriminant validity (the pairs of different constructs) were considered. Thus, the correlations of the pre-test–post-test differences (i.e., representing the magnitude of change during the exercise) between FS and SAM-P were calculated for the valence dimension and between FAS and SAM-A for the arousal dimension (convergent validity). The correlations of the magnitude of change between the pairs of different constructs (e.g., FS and SAM-A; FAS and SAM-P) were also calculated for discriminant validity. To compare the present data to the findings of Maibach et al. [28], the average for the post-test, the magnitude of change, and the direction of change between the pre-test and post-test were considered. In addition, the number of zero variations were analyzed for all (sub)scales.

## 3. Results

### 3.1. Correlation of (Sub)Scales

The correlations between two (sub)scales testing the same construct (i.e., indicating convergent validity) and between two (sub)scales addressing different constructs (i.e., supporting discriminant validity) are presented in Table 1. For convergent validity, the FS and SAM-P (*r* = 0.50) and the FAS and SAM-A (*r* = 0.16) were positively correlated, as expected. According to Cohen [36], the correlation for the valence dimension is of a moderate effect size and, for the arousal dimension, the correlation is of a small effect size. For discriminant validity, the FS and FAS (*r* = −0.10), the FS and SAM-A (*r* = −0.13), the FS and SAM-D (*r* = 0.13), the FAS and SAM-P (*r* = −0.10), and the FAS and SAM-D (*r* = 0.13) did not correlate significantly, which was also expected. 

When statistically comparing the effect size by Fisher’s z-transformation of the significant correlations of the same constructs with the ones reported by Maibach et al. [28], the correlation coefficients were significantly smaller for the valence dimension (*z* = 2.63, *p* = 0.008) and for the arousal dimension (*z* = 2.90, *p* = 0.004).

### 3.2. Analysis of Affective Responses

The mean pre-test and post-test values (with standard deviation) and the ranges of the answers for the different (sub)scales are provided in Table 2. For the valence dimension, the mean of the affective response before the JG exercise (i.e., at the pre-test) was as follows: for the FS_pre_, 1.98 ± 1.68; for the SAM-P_pre_, 3.86 ± 0.58. The mean of the affective response after the JG exercise (i.e., at the post-test) was as follows: for the FS_post_, 2.87 ± 1.74; for the SAM-P_post_, 4.19 ± 0.61. The magnitude of change from pre-test to post-test was significant for the FS_change_ (at 0.95 [*t*(193) = 6.546, *p* < 0.001, *d* = 0.47]) and for the SAM-P_change_ (0.33 [*t*(193) = 6.419, *p* < 0.001, *d* = 0.46]). The number of participants displaying zero variations was 35 for the FS and 95 for the SAM-P (see Table 3). For the arousal dimension, the mean of the affective response after the JG exercise (i.e., at the pre-test) was as follows: for the FAS_pre_, 2.92 ± 1.12; for the SAM-A_pre_, 3.18 ± 1.03. The mean of the affective response after the JG exercise (i.e., at the post-test) was as follows: for the FAS_post_, 3.25 ± 1.41; for the SAM-A_post_, 3.18 ± 1.13. The magnitude of change from pre-test to post-test was significant for the FAS_change_ (0.34 [*t*(193) = 3.118, *p* < 0.05, *d* = 0.22]) but not for the SAM-A_change_ (−0.01). The number of participants displaying zero variations was 42 for the FAS and 79 for the SAM-A (see Table 3). The magnitude of change from pre-test to post-test was significant for the dominance dimension (SAM-D_change_ = 0.29 [*t*(193) = 4.638, *p* < 0.001, *d* = 0.33]). The number of participants displaying zero variations was 96 (see Table 3).

## 4. Discussion

Recently, Maibach et al. [28] published a validation of the German versions of the FS and the FAS that examined the affective responses for a high-intensity exercise (i.e., the bicycle ergometer task). In the present study, these two questionnaires were supplemented by the SAM [29] to examine the affective responses for a JG exercise performed at a constant level of moderate intensity. To this end, participants performed the JG exercise for 45 min, and the questionnaires were completed in a pre-test-intervention-post-test design. The results of the present study demonstrate that the German versions of the FS and the FAS by Maibach et al. [28] are also effective for measuring the affective responses after moderate-intensity exercises (here, JG).

### 4.1. Analysis of Correlation

For the proof of validity, the discussion follows the interpretation of multitrait-multimethod (MT-MM) analysis [35,37]. When looking at convergent validity (i.e., correlation of two pairs from the same construct measured by different questionnaire methods), the results of the correlation coefficient of the valence dimension between the FS and the SAM-P is significant with a moderate effect size, and the correlation coefficient of the arousal dimension between the FAS and the subscale SAM-A is significant with a small effect size. Thus, the convergent validity can be assumed for the FS and the FAS (Hypothesis 1). For discriminant validity, three criteria must be met, according to MT-MM analysis [35]: first, the correlation for pairs of different constructs measured by the same questionnaire method (i.e., numerical items [FS and FAS] or pictorial items [SAM-P and SAM-A]) must be lower than the correlation for the pairs of the same construct measured by different questionnaire methods (e.g., FS and SAM-P, FAS and SAM-A). Second, the correlation for pairs of different constructs measured by different questionnaire methods (FS and SAM-A, FS and SAM-D, FAS and SAM-P, FAS and SAM-D) must be lower than the correlation for pairs of the same construct (FS and SAM-P, FAS and SAM-A). Third, the correlations must follow the same pattern, e.g., the same direction of the correlation coefficient (plus or minus) for the same pairs of constructs. All three criteria are met for the scales of the valence dimension and for the scales of the arousal dimension, providing support for their discriminant validity (Hypothesis 2). Together, these results confirm the construct validity of the German translations of the FS and the FAS by Maibach et al. [28] for the moderate-intensity JG exercise in the present study.

### 4.2. Variability of Answers

The comparison of the mean values of the affective responses after the JG exercise of the FS with the study of Maibach et al. [28] confirms Hypothesis 3, which stated that the post-test mean of the moderate-intensity exercise (here JG, FS_post_ = 2.87) should be higher than the post-test mean after a high-intensity exercise (bicycle ergometer task, FS_post_ = −0.29). This finding is in line with previous assumptions [6,9,13]. In addition, the post-test mean value of the FS in the present study is comparable with the ones of a previous study by Van Landuyt et al. [38], who used a moderate-intensity bicycle ergometer task, and of a study by Oliveira et al. [12], who used continuous training on a treadmill. The mean value is well within the range of the FS scores suggested by Bok et al. [39] for self-paced physical activities of moderate intensity (i.e., in the range of “feels good”—a score between 2.06 and 4.18). 

Hypothesis 4 is also confirmed. The mean of post-test values of the FAS in the present study (FAS_post_ = 3.25) is lower than in the study by Maibach et al. [28] (FAS_post_ = 4.73). This is in line with Ekkekakis et al. [6], who proposed higher arousal scores for high-intensity exercises as compared to moderate-intensity exercises. Also, like the FS scores, the FAS scores of the present study are similar to the ones gathered in the moderate-intensity bicycle ergometer task of Van Landuyt et al. [38] and the continuous treadmill task of Oliveira et al. [12].

As expected, the direction of change for the FS and for the FAS was positive in the present study. Within the circumplex model, this pattern of results can be mapped to the unactivated–pleasant affective state ([5]; Figure 1), which can be characterized by the experience of contentment [23]. Also, the magnitude of change for the FS and for the FAS shows a smaller range for the (relative) moderate-intensity JG exercise in the present study when compared to the stepwise increase in physical load to subjective exhaustion during the bicycle task in Maibach et al. [28]. In the present study, the magnitude of change between the pre-test and post-test was 3.15 points smaller for the FS and 1.92 points smaller for the FAS than in Maibach et al. [28]. Accordingly, Hypothesis 5 and Hypothesis 6 can be accepted. It should be noted here that the change in the FS and the FAS in the study by Maibach et al. [28] is calculated between the first measurement during the bicycle ergometer task (i.e., after the first load level at 60 watts for men and at 50 watts for women) and the post-test value after exercise. Thus, it can be assumed that the magnitude of change for the FS is somewhat greater still, since an increase in valence can be expected even at a low intensity level [6,39]. In any event, the magnitude and pattern of changes for the FS and the FAS in the present study is similar to what can be derived from the literature for moderate-intensity exercises (e.g., [6,38,39]).

Due to the (relatively) constant level of moderate intensity during the JG exercise, a higher proportion of zero variations was expected for the FAS as compared to the increased effort during the bicycle ergometer task in Maibach et al. [28]. In the study of Maibach et al. [28], only 10 of 82 participants (12.2%) displayed no variation from pre-test to post-test, whereas in the present study, 42 of 194 participants (21.6%) responded without variations. Thus, Hypothesis 7 can be confirmed. The percentage of zero variations in the present study is further consistent with the results found for the moderate-intensity bicycle ergometer task of Van Landuyt et al. [38]. Being instructed to choose a self-selected, preferred speed at which they felt pleasant gave the participants the ability to control the exercise intensity and the related perceived activation [18].

A limitation of the present study is that the perceived exertion during or after the JG exercise was not assessed with a separate scale, such as the rating of perceived exertion (RPE) scale [30], to capture the subjectively perceived exertion of the participants and, thus, to better monitor the moderate level of the exercise’s intensity. Simply instructing participants to run at a self-selected, preferred speed, at which they felt pleasant, without heavy breathing and being able to talk to each other during the run, might not have been sufficient to ensure that participant always maintained a moderate intensity level, even when being accompanied by an experience running instructor, who provided extra supervision, during the exercise. Also, the high number of zero variations for the SAM scales may be seen as a problem because it suggests that a large number of participants did not respond to the 45 min exercise intervention (at least, as measured with the SAM). To reduce the number of zero variations for the subscales of the SAM, a nine-point rating scale could have been used to capture minor changes in the affective responses (as had been already described as an alternative by Bradley and Lang [29]) which were otherwise not captured with the five-point rating scale.

## 5. Conclusions

The results of the present study demonstrate that the German versions of the FS and the FAS can be also used to examine the affective responses for a jogging exercise performed at a constant level of moderate intensity. The correlation analysis confirmed the assumption of the convergent and the discriminant validity; thus, the conditions for the construct validity are met. In addition, the present results supported all hypotheses for young adults (18–35 years old) derived from the comparison with the previous study by Maibach et al. [28], who tested middle-aged adults (35–65 years old). If the affective responses (affective valence and arousal) are transferred to the circumplex model [5], the participants after (post) a high-intensity bicycle ergometer task are, on average, in the activated-unpleasant quadrant and the participants after (post) a moderate-intensity jogging exercise are, on average, in the unactivated-pleasant quadrant. As expected, the values after the moderate-intensity jogging exercise are higher for the FS and lower for the FAS compared to the high-intensity bicycle ergometer task. In addition, a smaller magnitude of change for the FS and the FAS and a higher percentage for zero variations for the FAS was observed for the jogging exercise.

In the future, it should be investigated if the German versions of the two single-item scales generalize to different populations such as adolescents (13–17 years), older adults (65–80 years), or different clinical populations. The FS and the FAS can be helpful diagnostic tools to efficiently find the optimal exercise intensity for each participant within an exercise program without, for example, measuring heart rate or other physiological parameters. At the same time, these two single-item questionnaires can be used to compare the expected effect (e.g., increase in affective valence, decrease in arousal) of the exercise chosen with the actual effect and to determine the individually perceived benefits of leisure time activities [7]. Finding the right exercise that matches the preferences and the optimal intensity level that results in pleasant affective experiences can support adherence to physical activity and active living (ART, [4]).

## Figures and Tables

**Figure 1 behavsci-14-00317-f001:**
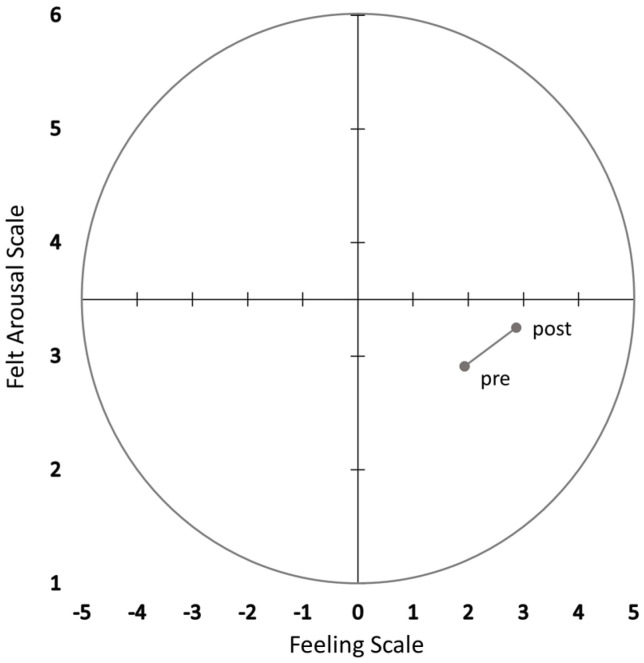
Affective response before (pre) and after (post) the jogging (JG) exercise in accordance with the two-dimensional model by Ekkekakis and Petruzzello [5].

**Table 1 behavsci-14-00317-t001:** Correlations between the different (sub)scales for the magnitude of change between pre-test and post-test (N = 194).

Variables	FS	FAS	SAM-P	SAM-A	SAM-D
FS	-				
FAS	−0.10	-			
SAM-P	0.50 **	−0.10	-		
SAM-A	−0.13	0.16 *	−0.07	-	
SAM-D	0.13	0.13	0.18 *	0.23 **	-

Note: FS = Feeling Scale, FAS = Felt Arousal Scale, SAM-P = pleasure dimension of Self-Assessment Manikin, SAM-A = arousal dimension of Self-Assessment Manikin, SAM-D = dominance of Self-Assessment Manikin. * The correlation is significant at 0.05 (two-sided). ** The correlation is significant at 0.01 (two-sided).

**Table 2 behavsci-14-00317-t002:** Descriptive statistics of the pre-test and post-test data (N = 194).

Scales Used	Range of the Scale	Mean Values	Standard Deviations	Range of Answers
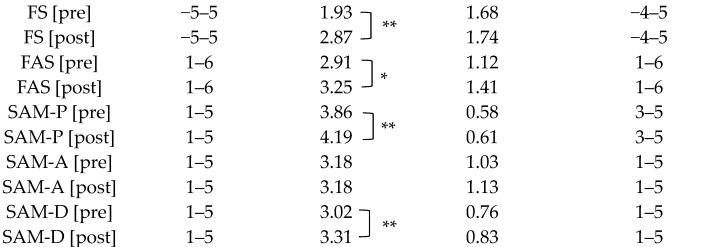				

Note: FS = Feeling Scale, FAS = Felt Arousal Scale, SAM-P = pleasure dimension of the Self-Assessment Manikin, SAM-A = arousal dimension of the Self-Assessment Manikin, SAM-D = dominance of Self-Assessment Manikin. Significant differences from pre-test to post-test are indicated as follows: * Significant at 0.05. ** Significant at 0.01.

**Table 3 behavsci-14-00317-t003:** Descriptive statistics of the zero variations and the direction of change from pre-test to post-test (N = 194).

Variables	Zero Variation n (in %)	Increase n (in %)	Decrease n (in %)
FS	35 (18.0)	132 (68.1)	27 (13.9)
FAS	42 (21.6)	97 (50.0)	55 (28.3)
SAM-P	95 (49.0)	80 (41.3)	19 (9.8)
SAM-A	79 (40.7)	61 (31.4)	54 (27.8)
SAM-D	96 (49.5)	73 (37.6)	25 (12.9)

Note: FS = Feeling Scale, FAS = Felt Arousal Scale, SAM-P = pleasure dimension of the Self-Assessment Manikin, SAM-A = arousal dimension of the Self-Assessment Manikin, SAM-D = dominance of Self-Assessment Manikin.

## Data Availability

The publicly archived dataset generated during the study can be downloaded at the following link: https://osf.io/pk3cb (accessed on 1 April 2024).

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
