# Peer review of "The Affective Responses to Moderate Physical Activity: A Further Study to Prove the Convergent and the Discriminant Validity for the German Versions of the Feeling Scale and the Felt Arousal Scale"

_behavsci, 2024, doi:10.3390/bs14040317_

Round 1

Reviewer 1 Report

Comments and Suggestions for Authors

Article: The affective responses to moderate physical activity: A further 2 validation study for the German versions of the Feeling Scale 3 and the Felt Arousal Scale

Aspects to improve:

1. Abstract: Ensure that the abstract offers a clear and concise overview of the purpose, methods, main results and conclusions of the study.

2. Introduction: Ensure that the introduction incorporates current studies related to the object of study, clearly establishes the context, the relevance of the study and specifically defines the objectives.

3. Materials and Methods: Present a detailed description of the methodology (design, type of research) that allows the replicability of the study, including an adequate justification of the choice of instruments and participants (what were the inclusion and exclusion criteria).

4. Results: Ensure that results are presented clearly, with appropriate statistical analysis and accurate interpretation of data.

5. Discussion: Deepen the interpretation of the results, relating them to previous studies and discussing the theoretical and practical implications, as well as the limitations of the study.

6. Conclusions: Ensure that conclusions follow logically from the results and offer recommendations based on the study findings.

7. References: Verify that all references are current, relevant and adequately cited to support the study and its claims. Likewise, there are documents from 1980 that are very old. Update from the last 5 years.

Reviewer 2 Report

Comments and Suggestions for Authors

Sorry because my writing in English is not optimal.

My specialty is methodology and statistics and from where I make comments.

I recommend that the title describe the type of validity study carried out: convergent and discriminant.

The introduction is correct: there are the most relevant previous studies. In the text there is only one study on these two scales in the years 2021, 2022 and 2023. Are there none more?

The objective is clear, and so are the hypotheses.

The sample, instruments and procedure are well described.

The statistics used are correct.

I suggest from table 1 delete the first row (FS) and the last column (SAM-D).

I suggest including in Table 2 or Table 3 some indicator of significance of the differences found.

In section 3.2 of results, only the means for the post moment are given; it would be good to also include those for the pre moment.

The conclusions are correct

Reviewer 3 Report

Comments and Suggestions for Authors

The paper submitted for review concerns the validation of German versions of the Feelings and the Felt Arousal Scales. The paper includes an extensive and comprehensive literature review. The methodological part is not very objectionable however I will refer to it in the minor comments. The results are presented in a clear manner however the statistical methods used could have been more insightful. The discussion is extensive and concludes with a paragraph of limitation where the authors point out the weaknesses of the paper. The conclusions should correspond in more detail with the hypotheses. 

Minor comments:

 - the paragraph with the purpose of the work is too elaborate, some of the information there is about material and methods. I suggest that you think about rewriting this section.

- The statistical analysis is devoid of methods of inductive statistics, except for correlation and Student's test (I suppose so because I did not find the information) The authors could use more sophisticated methods. 

- The authors formulated 7 hypotheses for the purpose, I would ask them to address the hypotheses in more detail in the conclusions.

Finally, I would like to congratulate the Authors on their good work.

Round 2

Reviewer 1 Report

Comments and Suggestions for Authors

Article: The affective responses to moderate physical activity: A further 2 validation study for the German versions of the Feeling Scale 3 and the Felt Arousal Scale

Aspects to improve:

1. Summary: Ensure that the summary provides a clear and concise description of the purpose, methods, main results, and conclusions of the study.

2. Introduction: Ensure that the introduction incorporates current studies related to the object of study, clearly establishes the context, the relevance of the study and specifically defines the objectives.

3. Materials and Methods: Present a detailed description of the methodology (design, type of research) that allows the replicability of the study, including an adequate justification of the choice of instruments and participants (what were the inclusion and exclusion criteria).

4. Results: Ensure that the results are presented clearly, with appropriate statistical analysis and accurate interpretation of the data. In a more didactic way, it should be done by objective or by stated hypothesis.

5. Discussion: Deepen the interpretation of the results, relating them to previous studies and discussing the theoretical and practical implications, as well as the limitations of the study.

6. Conclusions: Ensure that conclusions follow logically from the results and offer recommendations based on the study findings.

7. References: If they have incorporated updated bibliography, documents that are more than 20 years old are still evident.

The authors have not made all the suggested changes, the suggestions made are still maintained.

Round 3

Reviewer 1 Report

Comments and Suggestions for Authors

Article: The affective responses to moderate physical activity: A further 2 validation study for the German versions of the Feeling Scale 3 and the Felt Arousal Scale

Aspects to improve:

1. Summary: has been improved

2. Introduction: updated studies have been incorporated

3. Materials and Methods: Solvented

4. Results: Solved

5. Discussion: Solved

6. Conclusions: Incorporate the limitations of the study.

7. References: If they have incorporated updated bibliography.

The authors have modified what was requested